# The Influence of *Hop Latent Viroid* (HLVd) Infection on Gene Expression and Secondary Metabolite Contents in Hop (*Humulus lupulus* L.) Glandular Trichomes

**DOI:** 10.3390/plants10112297

**Published:** 2021-10-26

**Authors:** Josef Patzak, Alena Henychová, Karel Krofta, Petr Svoboda, Ivana Malířová

**Affiliations:** Hop Research Institute Co., Ltd., Kadaňská 2525, 438 01 Žatec, Czech Republic; henychova@chizatec.cz (A.H.); krofta@chizatec.cz (K.K.); svoboda@chizatec.cz (P.S.); malirova@chizatec.cz (I.M.)

**Keywords:** hop, *Humulus lupulus*, hop latent viroid, HLVd, bitter acids content, xanthohumol, essential oils, differential gene expression

## Abstract

Viroids are small infectious pathogens, composed of a short single-stranded circular RNA. Hop (*Humulus lupulus* L.) plants are hosts to four viroids from the family *Pospiviroidae*. Hop latent viroid (HLVd) is spread worldwide in all hop-growing regions without any visible symptoms on infected hop plants. In this study, we evaluated the influence of HLVd infection on the content and the composition of secondary metabolites in maturated hop cones, together with gene expression analyses of involved biosynthesis and regulation genes for Saaz, Sládek, Premiant and Agnus cultivars. We confirmed that the contents of alpha bitter acids were significantly reduced in the range from 8.8% to 34% by viroid infection. New, we found that viroid infection significantly reduced the contents of xanthohumol in the range from 3.9% to 23.5%. In essential oils of Saaz cultivar, the contents of monoterpenes, terpene epoxides and terpene alcohols were increased, but the contents of sesquiterpenes and terpene ketones were decreased. Secondary metabolites changes were supported by gene expression analyses, except essential oils. Last-step biosynthesis enzyme genes, namely humulone synthase 1 (HS1) and 2 (HS2) for alpha bitter acids and O-methytransferase 1 (OMT1) for xanthohumol, were down-regulated by viroid infection. We found that the expression of ribosomal protein L5 (RPL5) RPL5 and the splicing of transcription factor IIIA-7ZF were affected by viroid infection and a disbalance in proteosynthesis can influence transcriptions of biosynthesis and regulatory genes involved in of secondary metabolites biosynthesis. We suppose that RPL5/TFIIIA-7ZF regulatory cascade can be involved in HLVd replication as for other viroids of the family *Pospiviroidae*.

## 1. Introduction

Viroids are non-encapsidated, covalently closed, non-coding circular RNA molecules consisting of 246 to 399 nucleotides. Viroid species are phylogenetically classified into two families: *Pospiviroidae* and *Avsunviroidae*. Viroids of the family *Pospiviroidae* are replicated through an asymmetric rolling-circle mechanism in the nucleus using host DNA-dependent RNA polymerase II [1]. Hop (*Humulus lupulus* L.) plants are hosts to several viroids from the family *Pospiviroidae*. Originally, hop stunt viroid (HSVd), of the genus *Hostuviroid*, and hop latent viroid (HLVd), of the genus *Cocadviroid*, have occurred in hop [2]. Recently, apple fruit crinkle viroid (AFCVd), of the genus Apscarviroid [3], and citrus bark cracking viroid (CBCVd), of the genus *Cocadviroid* [4], have been found in hop plants. AFCVd and HSVd infection induces symptoms, which include stunting, leaf curling, small cone formation, and a considerable reduction of alpha bitter acid content in cones [3,5]. Bitter acids, which are essential compounds of beer bitterness, are prenylated polyketides synthesized in lupulin glands and divided to alpha and beta with one more prenylation step in synthesis [6]. CBCVd infection causes similar, more severe symptoms, which often lead to plant death [7,8]. The spreading of these viroids has been limited to local outbreaks but HLVd infection has been reported worldwide in hop-growing regions [2,7,9,10].

Although HLVd-infected hop plants are symptomless, infection leads to a significant reduction in cone yield and bitter acids content in hop cones [11]. The yield was lower, from 8% to 35%, for infected plants of the cultivars Wye Challenger and Omega. The content of alpha bitter acids was reduced by 15% and 30%, whie the content of beta bitter acids was slightly higher. Follow-up experiments with Wye Challenger showed an 11% reduction of the yield, 11% reduction of the content of alpha bitter acids, and 8% increase of the content of beta bitter acids [12]. The reduction of alpha bitter acids’ content due to HLVd infection ranged from 20% to 50% within English hop cultivars and was genotype-dependent [13]. Similar results were found for hop cultivars Saaz, Premiant (40% reduction) [9]; Aurora (18% reduction) [8]; and Sybillla, Marynka, Pulawski, and Magnat (from 11% to 23% reduction) [14]. The content of beta bitter acids was slightly higher (0–5%) for all the cultivars. The loss of yield due to HLVd infection reached from 15% to 37.5% for Slovenian hop cultivars [8] and from 6.4% to 15.3% for Polish hop cultivars [14]. Viroid infection also influences the essential oils composition in hop cone, which contributes to the aroma flavor of beer. The increase of myrcene content by 38% for Wye Challenger-infected plants was first reported [12]. Similar results were found for hop cultivars Saaz and Premiant when the content of myrcene was increased by 29% together with monoterpene pinene isomers (about a 40% increase) for infected plants. On the contrary, all sesquiterpenes were reduced by 4.4% to 29% in cones of infected plants. From other compounds, terpene alcohols (linalool, geraniol, and methylgeranate) and epoxides were increased and ketones were decreased for infected plants [9]. Therefore, the composition of essential oils in hop cones is genotype-dependent and specific [15], and these changes cannot be general. Trends for the content of sesquiterpenes and monoterpenes (myrcene and β-pinene) were similar within Polish hop cultivars, with the exception of myrcene for cultivar Sybilla [14]. The content of linalool was higher for cultivars Sybillla, Lubelski, and Pulawski, but lower for cultivars Marynka and Magnat in cones of infected plants. The content of methylgeranate was lower for infected plants of all cultivars.

Hop secondary metabolites are biosynthesized and accumulated in glandular trichomes, lupulin glands, and on the inner side of cone bracteoles and bracts. In the last two decades, numerous transcriptome and proteome studies have provided a systematic understanding of secondary metabolite biosynthesis pathways and elucidated the role of structural and regulatory genes for bitter acids [6,16,17,18,19,20,21,22], essential oils [23,24,25], and prenylated flavonoids [26,27,28,29,30,31,32] biosynthesis in hop. The high-throughput next-generation sequencing (NGS) technologies and computational biology tools allow to obtain rapid and cost-effective transcriptomic resources, providing better insight into viroid-hop plant interactions for understanding the host response to viroid infection [33,34,35]. The molecular mechanism of viroid-induced pathogenesis concerns the accumulated viroid-specific small RNAs (vd-sRNA) that are involved in transcriptional and post-transcriptional gene silencing via gene methylation and RNA interference (RNAi), and/or direct interaction with plant proteins [35]. In silico prediction of vd-sRNA targets found 1,062 unique targets for HLVd vd-sRNAs action [33]. Transcriptome analyses have shown the dynamic modulation of genes involved in protein, sugar metabolism, photosynthesis, physiology, phytohormone-signaling pathways, plant defense responses, and the cell wall structure. HLVd viroid infection has influenced an expression of plant resistance genes [33], transcription mediator genes [34], as well as transcription factors genes and secondary metabolites biosynthesis genes [35]. Changes of structural and regulatory genes for secondary metabolites biosynthesis were studied in leaves but the expression of these genes is the highest in lupulin glands inside hop cone.

Proteosynthesis machinery includes the regulation of several processes connected to 5S rRNA biogenesis by the ribosomal protein L5 (RPL5) and transcription factor IIIA (TFIIIA). TFIIIA protein expression is controlled by alternative splicing of the exon containing the plant 5S rRNA mimic (P5SM) [36]. Recently, a splicing variant, specifically TFIIIA-7ZF, was identified as essential for the replication of potato spindle tuber viroid (PSTVd) [37,38,39]. PSTVd modulates its expression through a direct interaction with RPL5, resulting in an optimized expression of TFIIIA-7ZF. It was found that TFIIIA-7ZF and RPL5 cloned from *Nicotiana benthamiana* also bind to HSVd [39]. It is suggested that the RPL5/TFIIIA-7ZF regulatory cascade is employed for replication by other viroids in the family *Pospiviroidae*; this was proved for AFCVd and CBCVd in infected tobacco plants. TFIIIA and RPL5 mRNA levels during the development of tobacco pollen are very low at late stages of mature and germinating pollen [40]. Viroid replication was depressed to lead to viroid elimination. HLVd is also non-transmissible via generative phases [41] and the RPL5/TFIIIA-7ZF regulatory cascade can similarly regulate its replication. It was found that TFIIIA-7ZF co-expression significantly influenced the tripartite Myb2/bHLH2/WDR1 [29] and bipartite WRKY1/WDR1 [30] complexes in *Nicotiana benthamiana*, activating hop chalcone synthases as well as O-methyltransferase and WRKY promoters [40,42]. These results suggest that TFIIIA-7ZF has the potential to modulate or change the expression of secondary metabolite genes in hop upon viroid infection.

In the present work, we aimed to evaluate the influence of HLVd infection on the content and composition of secondary metabolites in maturated hop cones. In parallel, we evaluated the expression levels of genes involved in secondary metabolites biosynthesis and regulation to provide a complex understanding of transcriptomic and metabolic changes caused by viroid infection.

## 2. Results

### 2.1. Contents of Secondary Metabolites in Maturated Hop Cones of Healthy and HVLd-Infected Plants

Total alpha and beta bitter acids, cohumulone and colupulone, and xanthohumol contents were determined by HPLC in dry cones of healthy and HLVd-infected plants of the Czech hop cultivars. The infectious status of each sample was checked before the analyses. The results of two harvest years (2019 and 2020) for cultivar Saaz are summarized in Table 1. The content of alpha bitter acids was significantly reduced for both years by 30.6% and 34.2%, respectively, in infected hop plants (Figure 1A). The content of beta bitter acids was not influenced by viroid infection and its variation was due to environmental factors of harvest years. Similar results were obtained for cohumulone and colupulone, as well as for bitter acids homologues’ contents (Table 1). The content of xanthohumol was significantly reduced for both years by 21.7% and 23.5%, respectively, in infected hop plants (Table 1). The results of harvest year 2020 for other cultivars, including Sládek, Premiant, and Agnus, are summarized in Table 2. Identical trends were found for contents of alpha bitter acids and xanthohumol. The content of alpha bitter acids was significantly reduced by 28.2%, 14.8%, and 8.8% in the infected hop plants of the studied cultivars (Figure 1B). The content of xanthohumol was also significantly reduced by 22.9%, 21.1%, and 3.9% in the infected hop plants of the studied cultivars (Table 2). We found a significant increasing of beta bitter acids content (15.9%) and a significant decreasing of cohumulone content (6.6%) for infected hop plants of cultivar Sládek (Table 2).

Next, compositions of essential oils (EO) were determined by GC in dry cones of healthy and HLVd-infected plants of Czech hop cultivars. The results of two harvest years (2019 and 2020) for cultivar Saaz are summarized in Table 3. The content of total essential oils was not significantly influenced by viroid infection. However, significant changes in the composition of essential oils were found. The contents of the studied monoterpenes in EO significantly increased in infected hop plants for both years. Myrcene content was higher by 24.6% and 11.1%, and the increasing α and β-pinene contents varied by 32.8% to 56.6% in infected hop plants for both years (Table 3). The contents of the individual sesquiterpenes in EO were significantly reduced in a range from 0% to 20% in infected hop plants for both years (Table 3). The contents of terpene alcohols in EO were significantly increased, specifically linalool by 50.7% and 64.4%, as well as geraniol with an increase of 6.2 and 2.5 times in the infected hop plants for both years (Table 3). Similarly, the contents of terpene epoxides in EO were increased in infected hop plants depending on the harvest year and the same applied vice versa, wherein contents of terpene ketones in EO were reduced in infected hop plants depending on the harvest year (Table 3). The situation inside the terpene esters group was variable if the contents of methylgeranate, methylheptanoate, methylnon-6-enoate, and methyldeca-4,8-dienoate were increased; however, the contents of methyloctanoate, methyl-8-methylnonanoate, methyldecanoate, and methyldodeca-3,6-dienoate were reduced in infected hop plants (Table 3). The results of the EO analyses for cultivars Sládek, Premiant, and Agnus are summarized in Table 4. The same trends as those in Saaz hop samples were not found and the contents of individual EO compounds non-significantly varied due to cultivar genome and environmental factors of the harvest years. The content of total essential oils was slightly reduced by viroid infection for all the cultivars (Table 4). Only the contents of linalool, geraniol, and methylgeranate were similarly increased in infected hop plants depending on the harvest year (Table 4).

Yields of dry hop cones were measured only for healthy and HLVd-infected plants of cultivar Saaz. The yield was significantly reduced for both years by 19.3% and 18%, respectively, in infected hop plants. The average yields were 0.659 ± 0.194 ** kg of DW per healthy plant and 0.532 ± 0.218 kg of DW per infected plant in 2019, and 0.672 ± 0.231 * kg of DW per healthy plant and 0.551 ± 0.227 kg of DW per infected plant in 2020.

### 2.2. Analyses of Gene Expressions in Hop Cones of Healthy and HVLd-Infected Plants

Relative expressions of secondary metabolites biosynthesis and regulatory genes were evaluated by qRT-PCR in young cones of healthy and HLVd-infected plants of Czech hop cultivars. Firstly, relative expressions of six bitter acid biosynthetic pathway enzyme genes (Table 5) were analyzed. Both genes BCAT1 (Figure 2A) and VPS (Figure 2B), which are enzymes for bitter acid precursors, were down-regulated by viroid infection in young cones of the Saaz (3.6 and 2.9 times), Sládek (2.2 and 1.7 times), and Premiant (1.3 and 1.4 times) cultivars. Differences in the relative expression of PT1L (Figurce 2C), which is also involved in prenylflavonoids biosynthesis, was found for infected plants of the Saaz (2 times lower expression) and Sládek (3.3 times lower expression) cultivars. The PT2 (Figure 2D) gene was only down-regulated by infection in cones of the Sládek (3.7 times) cultivar. Both genes HS1 (Figure 2E) and HS2 (Figure 2F), which are last-step enzymes of alpha bitter acids biosynthesis, were down-regulated by infection in cones of all the cultivars: Saaz (8.3 and 8.1 times), Sládek (3.3 and 2.3 times), Premiant (3.3 and 2.5 times), and Agnus (2 and 1.5 times).

Next, relative expressions of four polyphenol and two flavonoid biosynthetic pathway enzyme genes (Table 5) were analyzed. From these genes, the PAL (Figure 3A) gene was only up-regulated by infection in cones of the Premiant (5.7 times) cultivar and the CHSH1 (Figure 3C) gene was only up-regulated by infection in cones of the Saaz (2.6 times) cultivar. The OMT1 (Figure 3D) gene, which is an enzyme for last-step xanthohumol biosynthesis, was significantly down-regulated by infection in cones of the Saaz (2.7 times) cultivar. There were no significant differences between healthy and viroid-infected plants for flavonoid biosynthetic pathway enzyme genes (Figure 4).

The group of hop essential oils includes the wide spectrum of terpenic compounds. Therefore, we studied relative expressions of eight monoterpene and sesquiterpene biosynthetic pathway enzyme genes (Table 5). The GPPS-SSU (Figure 5A) gene, which is a part of heterodimeric enzyme complex for monoterpenes precursor biosynthesis, was down-regulated by infection in cones of all the cultivars: Saaz (2.5 times), Sládek (2.8 times), Premiant (10 times), and Agnus (1.7 times). The FPPS (Figure 5B) gene, which is an enzyme for sesquiterpenes precursor biosynthesis, was significantly up-regulated by infection in cones of the Saaz (2.3 times) cultivar but non-significantly down-regulated by infection in cones of the Sládek (1.8 times), Premiant (1.6 times), and Agnus (1.4 times) cultivars. Both genes MTS1 (Figure 5C) and MTS2 (Figure 5D), which are enzymes for monoterpenes biosynthesis, were down-regulated by infection in cones of all the cultivars: Saaz (2.1 and 2.1 times), Sládek (1.4 and 0.9 times), Premiant (1.6 and 2.8 times), and Agnus (1.1 and 3 times). Additionally, both genes STS1 (Figure 5E) and STS2 (Figure 5F), which are enzymes for sesquiterpenes biosynthesis, were down-regulated by infection in cones of the Saaz (5.9 and 2.3 times), Sládek (2.2 and 13.8 times), and Agnus (2 and 3.3 times) cultivars. The TPS9 (Figure 5G) gene, which is a terpene biosynthesis enzyme with an unknown substrate, was significantly up-regulated by infection in cones of the Agnus (3.1 times) cultivar.

Secondary metabolite biosynthesis pathways are regulated by transcription factors’ network. Therefore, we studied relative expressions of six known transcription factors with influence on the transcription of biosynthetic pathway enzyme genes (Table 5). The MYB3 (Figure 6A) transcription factor was significantly down-regulated by infection in cones of the Sládek (4.4 times) and Premiant (5.9 times) cultivars. The MYB8 (Figure 6B) transcription factor was non-significantly down-regulated by infection in cones of the Saaz (1.5 times) and Sládek (16 times) cultivars. The MYB78 (Figure 6C) transcription factor was significantly down-regulated by infection in cones of the Premiant (6.5 times) cultivar. The bHLH2 (Figure 6D) transcription factor was non-significantly down-regulated by infection in cones of all the cultivars except Saaz. The bHLH4 (Figure 6E) transcription factor was significantly down-regulated by infection in cones of thhe Saaz (1.6 times) cultivar but significantly up-regulated by infection in cones of the Sládek (2.6 times) cultivar. The WRKY1 (Figure 6F) transcription factor was significantly down-regulated by infection in cones of the Sládek (1.9 times) and Premiant (2.7 times) cultivars. Considering that transcription factors are involved in the regulation of gene expression, we screened promoter sequences of biosynthetic pathway enzymes for binding sites of MYB, bHLH, and WRKY factors (Table 6). To a lesser extent, two and three cis-acting regulatory DNA elements were found for MYB and bHLH (MYC) or WRKY transcription factors, respectively, in the studied promoter sequences.

Transcription factor IIIA (TFIIIA) was found to be involved in viroid replication [37] and its expression is controlled by alternative splicing. We tried to quantify correctly and incorrectly spliced variants by qRT-PCR and semi-quantitative RT-PCR. Viroid infection significantly increased the number of incorrectly spliced variants in all the cultivars (Figure 7). Values were dependent on the used method when the relative portion of incorrectly spliced variants was increased by 2.9% or 5.9% for Saaz, 51.8% or 7.7% for Sládek, 16.1% or 4.1% for Premiant, and 24.8% or 5.2% for Agnus cultivars. Additionally, the ribosomal protein L5 (Figure 8) gene was non-significantly up-regulated by infection in cones of all the cultivars: Saaz (2.7 times), Sládek (4.3 times), Premiant (9.5 times), and Agnus (9.4 times).

## 3. Discussion

Even though, HLVd infection does not induce any visible symptoms, it has affected the content and composition of secondary metabolites in maturated hop cones [8,9,11,12,13,14]. The reduction of alpha bitter acids content was the most noticeable effect of viroid infection. The reduction ranged from 8.8% to 34.2% in the studied hop cultivars. We confirmed previous findings for cultivars Saaz and Premiant [9]. Similar results were found for English [11,12,13], Slovenian [8], and Polish [14] hop cultivars. On the contrary, the content of beta bitter acids was not influenced by viroid infection in all the studies. There was a small increase for Saaz (2019 samples) and Sládek cultivars, as well as for English [12] and Polish [14] hop cultivars. These changes can be connected to the content of both compound groups in lupulin glands [43], where the reduction of alpha bitter acids content shifts the alpha/beta ratio.

We found the novel significant reduction of xanthohumol content in cones of infected hop plants. The contents of this prenylflavonoid have not been studied before. However, it is known that xanthohumol content highly correlates with alpha bitter acids content [44,45,46] and it could be expected.

We also confirmed that viroid infection changed the composition of essential oils in hop cone. Either the increase of monoterpenes, terpene epoxides, and terpene alcohols content or the reduction of sesquiterpenes and terpene ketones content in cones of infected Saaz hop plants were consistent with previous results [9]. Previously, the contents of methylgeranate, methylnon-6-enoate, and methyldecanoate from terpene esters were only studied with similar trends. Change variations were dependent on common biosynthetic pathways of individual compounds. These changes were not found for other studied cultivars. Due to the genetically specific composition of essential oils for hop cultivars, the influence of viroid-infected plants infection may not be the same. For example, myrcene contents were increased in viroid-infected cones for Wye Challenger [12], Marynka, Lubelski, Pulawski, and Magnat cultivars, but not for the Sybilla cultivar [14]. The contents of linalool, geraniol, and methylgeranate were also increased in cones of infected hop plants for all the studied cultivars. Previously, the content of linalool was increased by infection for Sybillla, Lubelski, and Pulawski cultivars, but decreased for Marynka and Magnat cultivars [14]. However, the content of methylgeranate was conversely reduced by viroid infection for all the Polish cultivars.

We evaluated the yield of dry hop cones only for the Saaz cultivar. Its reduction by viroid infection was in accordance with previous results for English [11,12], Slovenian [8], and Polish [14] hop cultivars.

It can be assumed that all the secondary metabolite changes have a genetic background in differential gene expressions caused by viroid infection. That is why we evaluated relative expressions of known biosynthesis and regulatory genes. Genes for bitter acid precursors, namely BCAT1 [17] and VPS [16], were down-regulated in viroid-infected plants of the studied cultivars, except Agnus. However, viroid infection only reduced alpha bitter acids contents and beta acids contents were not influenced. We did not measure bitter acids precursors’ contents and their levels could be sufficient for the next step biosynthesis. Prenyltransferase genes PT1L [19,20] and PT2 [21] are common for bitter acids and prenylflavonoids biosynthesis, and we could not determine which compound contents were influenced by the down-regulation for Saaz and Sládek cultivars. Recently, we found that alpha bitter acids content in glandular trichomes of hop cone depends on last-step alpha bitter acids biosynthesis by an expression of humulone synthase genes [47]. Both HS1 and HS2 [22] genes were down-regulated by infection in cones of all the cultivars and it can be supposed that their lower expressions cause the reduction of alpha bitter acids content in viroid-infected hop cones.

There were no changes for prenylflavonoid biosynthetic pathway enzyme genes PAL, 4CL2, and CHSH1 by viroid infection in cones. We found up-regulation only for Premiant (PAL) and Saaz (CHSH1), which are not only involved in lupulin gland metabolites. Previously, up-regulations of PAL, 4CL2, and CHS2 genes in HLVd-infected leaves of cultivar Celeia were found [34]. We can suppose that viroid infection activates pathogen plant response reaction. Flavonoids and phenylpropanoids protect plants from abiotic and biological stresses, viruses [48], or fungal [49] pathogens. However, flavonols (F3H) and anthocyanidins (LAR) biosynthetic pathway genes were not influenced by viroid infection. The OMT1 gene [28], for a last step of xanthohumol biosynthesis, was only down-regulated by infection in cones of the Saaz cultivar, even though the content of xanthohumol was significantly reduced for all the cultivars.

The changes, which were found for relative expressions of terpenic biosynthetic pathway enzyme genes, did not correspond to the contents of essential oil compounds. The GPPS-SSU [25], MTS1, and MTS2 [24] enzyme genes of monoterpenes biosynthesis were down-regulated by infection in cones of the Saaz cultivar, despite the contents of monoterpenes being significantly increased. In a similar manner, the FPPS [23] enzyme gene for the biosynthesis of sesquiterpenes’ precursor, was significantly up-regulated by infection in cones of Saaz; however, the contents of sesquiterpenes were significantly reduced. Conversely, the STS1 and STS2 [24] enzyme genes of sesquiterpenes biosynthesis were down-regulated by infection in cones of the Saaz cultivar, similar to the sesquiterpenes content reduction. Some of these gene expression trends were similar for other cultivars, except for the FPPS enzyme gene. However, there were no significant differences within essential oils compositions for Sládek, Premiant, and Agnus cultivars. Gene expression changes can influence different essential oils compounds, which is genotype-dependent [15]. Between differentially expressed genes in HLVd-infected leaves of cultivar Celeia [34], the STS1, TPS9, and NES genes were up-regulated. In our study, TPS9 was up-regulated only in cones of the Agnus cultivar and the NES gene in cones of the Premiant cultivar. In contrast, these genes were down-regulated in cones of the Sládek cultivar by viroid infection. Terpenes, similar to flavonoids and phenylpropanoids, are involved in plant defense response [50], therefore viroid infection can activate their biosynthesis. It was previously found that viroid infection changed expressions of other genes involved in plant resistance [33].

Contents of secondary metabolites in glandular trichomes are not only influenced by structural biosynthesis genes but also by regulatory networks of involved transcription factors [29,30,31,32]. From the studied factors, the MYB and bHLH transcription factors were mostly down-regulated by infection in cones and these differences can cause expression differences for biosynthesis pathways genes. MYB and bHLH can regulate transcription separately or in the tripartite MBW (MYB/bHLH/WDR) complex [29]. Promoter cis-acting regulatory DNA element analysis showed that the MBW complex cannot regulate BCAT1, OMT1, FPPS, MTS2, and TPS9 genes. For MYB regulatory factors, the cis-acting regulatory DNA element MYB1AT was present in all the promoter sequences, except the FPPS gene, while others were missing irregularly in more sequences. The cis-acting regulatory DNA element for the bHLH (MYC) and WRKY transcription factors were present in all the studied promoter sequences. Biosynthesis pathway genes can thus be regulated together.

Even though regulatory transcription factors can influence an expression of several genes together, there were too many changes which could be caused by viroid-induced RNA interference [35]. Recently, the RPL5/TFIIIA-7ZF regulatory cascade was found to be necessary for the replication of several viroids [37,38,39,40,41,42]. Changes in splicing variants of TFIIIA-7ZF caused by viroids can disbalance a proteosynthesis in general and influence secondary metabolite biosynthesis [42]. We confirmed that the level of alternatively spliced variant TFIIIA-7ZF was increased by viroid infection. This variant is also induced by the RPL5 protein [38], whose gene expression was similarly higher in cones of infected plants. Our results were in accordance with the results for PSTVd infection [37,38] and a regulation of viroid replication by TFIIIA-7ZF can be generalized for the whole *Pospiviroidae* family.

## 4. Materials and Methods

### 4.1. Plant Materials and Hop Latent Viroid (HLVd) Detection

Analyzed hop plants were obtained from in vitro multi-shoot culture of mericlones [9,51,52]. Individual mericlones were derived and regenerated from meristems of selected field mother plants in a maintenance-breeding hop garden (Hop Research Institute, Co. Ltd., Zatec, Czech Republic) during the years 1988–1989 and 2014–2015 for Saaz, and during the years 1996–1997 and 2014–2015 for hybrid cultivars (Sládek, Premiant, and Agnus; Hong Kong). Hop latent viroid (HLVd) infection was analyzed by molecular dot blotting hybridization using a ^32^P[dCTP]-labelled cDNA HLVd probe [9]. The viroid level was quantified by means of the STORM PhosphorImager device and ImageQuant software (Molecular Dynamics, Chatsworth, CA, USA). RNA samples, isolated by PureLink™Plant RNA Reagent (ThermoFisher Scientific, Waltham, MA, USA) according to the protocol, were used for HLVd detection by real-time quantitative RT-PCR by the QuantiTect SYBR Green RT-PCR Kit (Qiagen, Hilden, Germany) according to Patzak et al. [53]. Fourteen mericlones of Saaz, two of Sládek, and one of both Premiant and Agnus cultivars were viroid-free. Sixteen mericlones of Saaz and one of each hybrid cultivar were derived from HLVd-infected mother plants without viroid elimination. Two to eight in vitro plants of the selected mericlones were acclimatized in a glasshouse and were well-rooted as well as transferred from field conditions in 2018 for Saaz and 2019 for the hybrid cultivars to experimental hop gardens at the Steknik farm of the Hop Research Institute in Zatec. All the hop plants were grown under standard agronomic conditions. For gene expression analyses, young cone samples were collected in August 2020. Samples were immediately frozen by liquid nitrogen and stored in a deep freezer (−80 °C) before analyses. For chemical analyses, a minimum of 500 g of mature cone samples were collected at the end of August and beginning of September 2019 and 2020, and were kiln-dried at 55 °C for 8 h to attain a target moisture content of 10%. One gram of each fresh sample was stored in a freezer for viroid infection analyses, similar to in vitro mericlones.

### 4.2. Chemical Analyses

The dried cone samples were used for the chemical analyses of hop resins, polyphenols, and essential oils. Hop resins and polyphenols were determined by liquid chromatography (HPLC) with a diode array detector (DAD) according to the modified EBC 7.7 method (Analytica EBC, 1998) on the Nucleosil C18 column (Macherey-Nagel, Düren, Germany), 5 μm, 250 mm × 4 mm, using a Shimadzu LC-20A (Shimadzu Europe GmbH, Duisburg, Germany) liquid chromatograph [54]. The flow rate of the mobile phase was 0.8 mL/min. The detection was carried out at wavelengths of 314 nm (hop resins) and 370 nm (polyphenols). Hop resins, alpha and beta acids, and xanthohumol were quantified by external calibration standards. Hop essential oils were estimated from vacuum-concentrated water-distilled samples by gas chromatography (GC) on a capillary column DB 5 (Chromservis, Prague, Czech Republic, 30 m × 0.25 mm × 0.25 μm film thickness) using a Varian 3400 gas chromatograph (Varian Inc., Palo Alto, CA, USA) combined with the aFinnigan ITD 800 mass detector (Thermo Scientific, Waltham, MA, USA) [47]. Compound identification was based on a comparison of GC retention indices and mass spectra with those of the authentic compounds. A semiquantitative evaluation of hop oils compositions was performed on the basis of peak areas of individual components and was expressed relatively to the total integrated area of all the substances involved. STATISTICA 8.0 CZ (StatSoft, Tulsa, OK, USA) was used for the evaluation of chemical analyses data by basic statistic functions. SigmaPlot for Windows v.10.0.0.54 (Systat Sowtware Inc., San Jose, CA, USA) was used for statistical group and *t*-test analyses.

### 4.3. Gene Expression Analyses

RNAs were isolated from frozen young hop cone samples (53 samples for Saaz and 8 samples each for the hybrid cultivars) using the PureLink™Plant RNA Reagent (ThermoFisher Scientific, Waltham, MA, USA) and purified by DNaseI treatment on columns (RNeasy Plant Mini Kit, Qiagen, Hilden, Germany) [29]. HLVd infection in RNA samples was analyzed by qRT-PCR, as mentioned above. RNA samples (5 μg) were reverse-transcribed by the oligo (dT)_18_ primer and First Strand cDNA Synthesis Kit (Roche Diagnostics, Mannheim, Germany) at 50 °C for 60 min to cDNA. NGS genome information [55,56,57] was used for the sequence searching of secondary metabolites biosynthesis and regulatory genes (Table 5), as well as of their promoters. Two thousand pair-base promoter sequences before the ATG-start codon were screened for motifs of plant cis-acting regulatory DNA elements (Table 6) in the database New PLACE (https://www.dna.affrc.go.jp/PLACE, accessed on 8 January 2007). RealTimeDesign software (LGC Biosearch Technologies, Petaluma, CA, USA) was used for the design of real time PCR primers (Appendix A), which were custom-synthesized by Generi Biotech (Hradec Králové, Czech Republic). A total of 2 μL of 50× diluted cDNA was used for a 20 μL PCR reaction with the iTaq universal SYBR green supermix (Bio-Rad Laboratories, Hercules, CA, USA) in a Real-time PCR cycler CFX Connect (Bio-Rad Laboratories, Hercules, CA, USA). Five reference genes (Table 5) were used for the normalization of samples [47]. The relative expression to reference genes was calculated by the “Delta-delta method” (RE = 2^−^^∆CT^). SigmaPlot for Windows v.10.0.0.54 (Systat Sowtware Inc., San Jose, CA, USA) was used for statistical group and *t*-test analyses.

Similar qRT-PCR analysis was used for the detection of splicing variants of transcription factor IIIA by PCR primer combinations F1+R and F2+R (Figure 9). A percentage of the incorrectly spliced variant (F2+R) to all the variants (F1+R) was calculated for statistical group and *t*-test analyses (SigmaPlot for Windows v.10.0.0.54, Systat Software Inc., San Jose, CA). The second approach to detect splicing variants was reverse-transcription PCR on RNA samples. The PCR primer combination F3+R (Figure 9) was used for RT-PCR reactions by the OneStep RT-PCR kit (Qiagen, Hilden, Germany) according to the protocol in a TOne thermocycler (Biometra, Goettingen, Germany). PCR amplification products were resolved via electrophoresis in horizontal 2% agarose (SeaKem LE, FMC Bioproducts, Philadelphia, PA, USA) gels, visualized by ethidium bromide-staining, and scanned by a CCD camera (Bio-Print CX4, Vilber Lourmat, Collégien, France) to PC. The molecular size of products was estimated by comparison with the pGEM DNA marker and 100 bp ladder (Promega, Madison, WI, USA). The volume and intensity of the correctly and incorrectly spliced variants were quantified by the quantification module of the Bio-Print CX4 software (Vilber Lourmat, Collégien, France). The percentage of the incorrectly spliced variant was used for statistical group and *t*-test analyses (SigmaPlot for Windows v.10.0.0.54, Systat Sowtware Inc., San Jose, CA, USA).

## 5. Conclusions

HLVd infection significantly affected the content and composition of secondary metabolites in maturated hop cones of the studied cultivars. The contents of alpha bitter acids and xanthohumol were significantly reduced in the infected hop plants of all the cultivars. The contents of monoterpenes, terpene epoxides, and terpene alcohols were increased, but the contents of sesquiterpenes and terpene ketones were decreased within essential oils in cones of infected Saaz hop plants. For this cultivar, the yield of the dry cones was also significantly reduced for both studied years.

Secondary metabolite changes caused by viroid infection were in accordance to the changes of the relative expressions of the studied biosynthesis and regulatory genes; this mainly concerned the down-regulation of last-step biosynthesis enzyme genes for alpha bitter acids (HS1 and HS2) and xanthohumol (OMT1) syntheses. The results for the essential oils biosynthesis genes were not so conclusive and sometimes contradictory.

Viroid infection significantly affected the expression and splicing of TFIIIA-7ZF mRNA, which can influence the proteosynthesis of the biosynthesis and regulatory genes involved in secondary metabolite biosynthesis. We suppose that the RPL5/TFIIIA-7ZF regulatory cascade can be required for the replication of HLVd, similarly to other viroids of the family *Pospiviroidae*.

## Figures and Tables

**Figure 1 plants-10-02297-f001:**
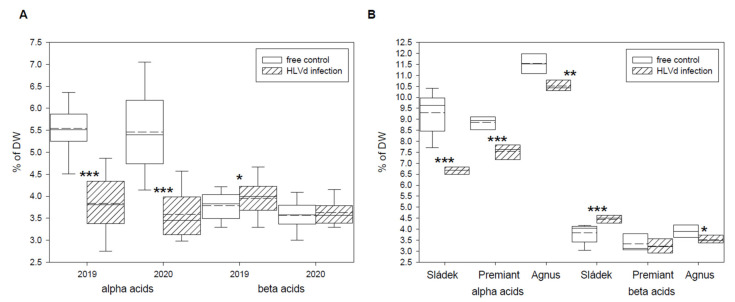
Contents of hop bitter acids in dry cones of HLVd-free and infected plants of the (**A**) Saaz and (**B**) hybrid cultivars. Probability level: *—*p* < 0.1, ******—*p* < 0.05, and ***—*p* < 0.01. Note: straight line—median; dashed line—average; and box—95% percentile ± standard deviation.

**Figure 2 plants-10-02297-f002:**
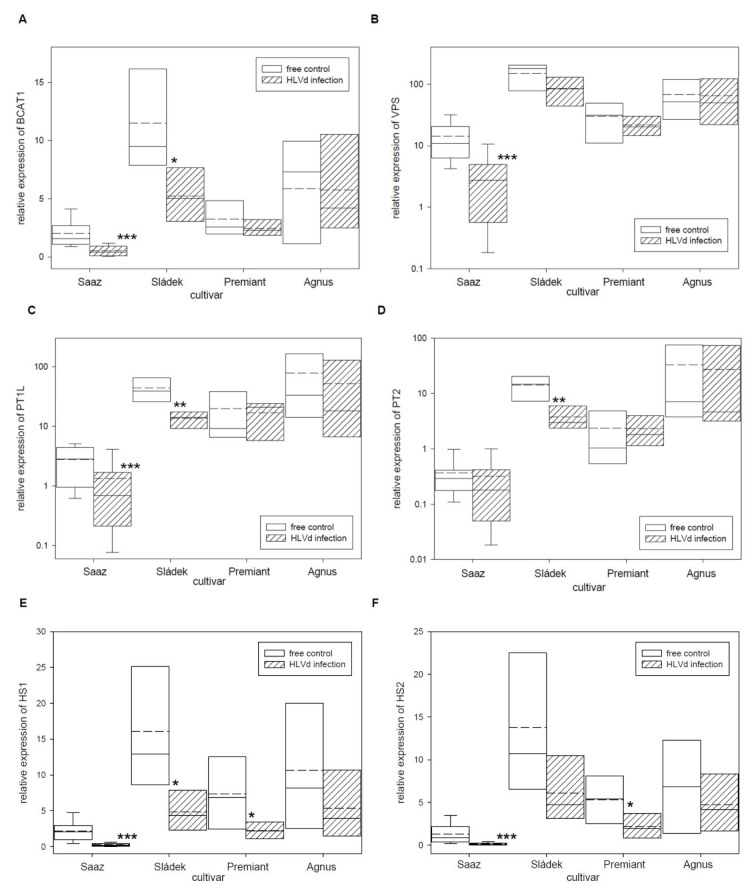
The relative expression to reference genes (RE = 1) for bitter acids biosynthesis genes (**A**) BCAT1, (**B**) VPS, (**C**) PT1L, (**D**) PT2, (**E**) HS1, and (**F**) HS2 in young cones of HLVd-free and infected plants of hop cultivars. Probability level: *—*p* < 0.1, ******—*p* < 0.05, and ***—*p* < 0.01. Note: straight line—median; dashed line—average; and box—95% percentile ± standard deviation.

**Figure 3 plants-10-02297-f003:**
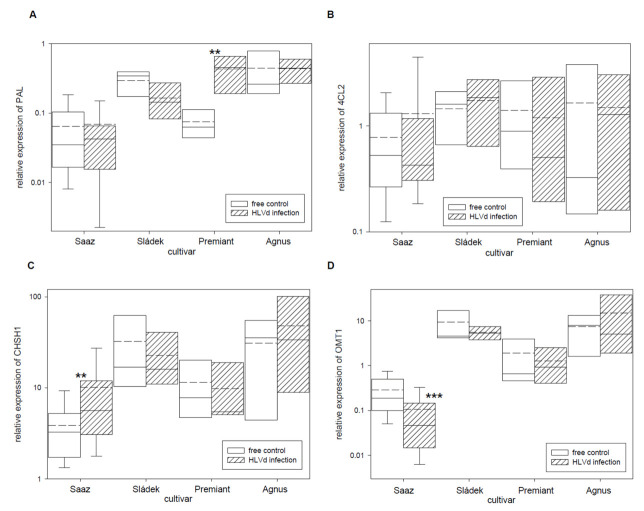
The relative expression to reference genes (RE = 1) for polyphenols biosynthesis genes (**A**) PAL, (**B**) 4CL2, (**C**) CHSH1, and (**D**) OMT1 in young cones of HLVd-free and infected plants of hop cultivars. Probability level: *—*p* < 0.1, ******—*p* < 0.05, and ***—*p* < 0.01. Note: straight line—median; dashed line—average; and box—95% percentile ± standard deviation.

**Figure 4 plants-10-02297-f004:**
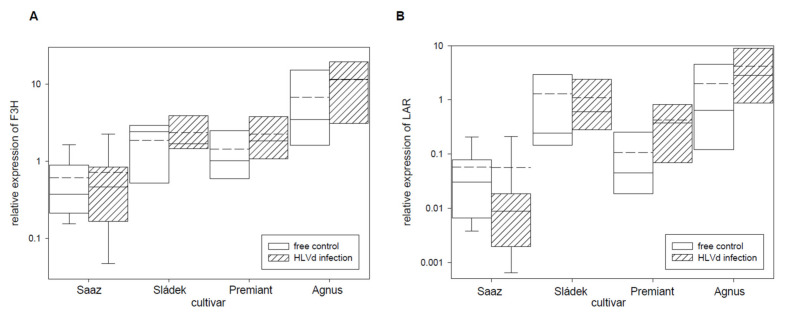
The relative expression to reference genes (RE = 1) for flavonoids biosynthesis genes (**A**) F3H and (**B**) LAR in young cones of HLVd-free and infected plants of hop cultivars. Probability level: *—*p* < 0.1, ******—*p* < 0.05, and ***—*p* < 0.01. Note: straight line—median; dashed line—average; and box—95% percentile ± standard deviation.

**Figure 5 plants-10-02297-f005:**
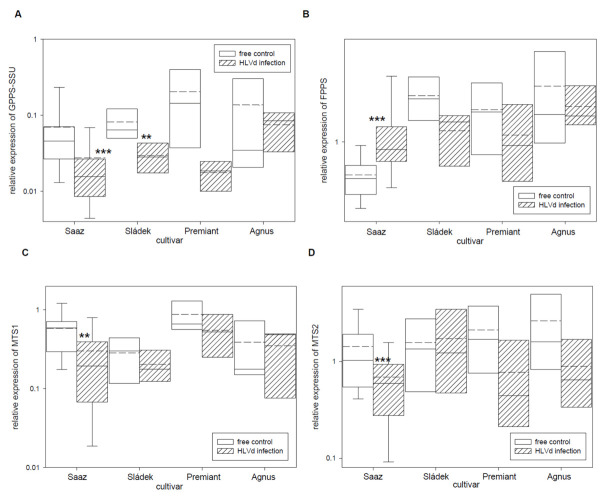
The relative expression to reference genes (RE = 1) for terpenes biosynthesis genes (**A**) GPPS-SSU, (**B**) FPPS, (**C**) MTS1, (**D**) MTS2, (**E**) STS1, (**F**) STS2, (**G**) TPS9, and (**H**) NES in young cones of HLVd-free and infected plants of hop cultivars. Probability level: *—*p* < 0.1, ******—*p* < 0.05, and ***—*p* < 0.01. Note: straight line—median; dashed line—average; and box—95% percentile ± standard deviation.

**Figure 6 plants-10-02297-f006:**
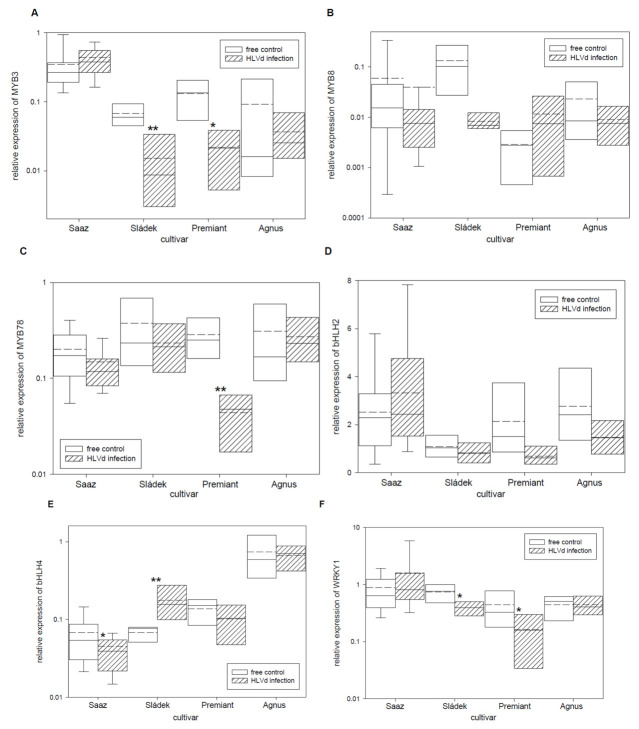
The relative expression to reference genes (RE = 1) for biosynthesis transcription factor genes (**A**) MYB3, (**B**) MYB8, (**C**) MYB78, (**D**) bHLH2, (**E**) bHLH4, and (**F**) WRKY1 in young cones of HLVd-free and infected plants of hop cultivars. Probability level: *—*p* < 0.1, ******—*p* < 0.05, and ***—*p* < 0.01. Note: straight line—median; dashed line—average; and box—95% percentile ± standard deviation.

**Figure 7 plants-10-02297-f007:**
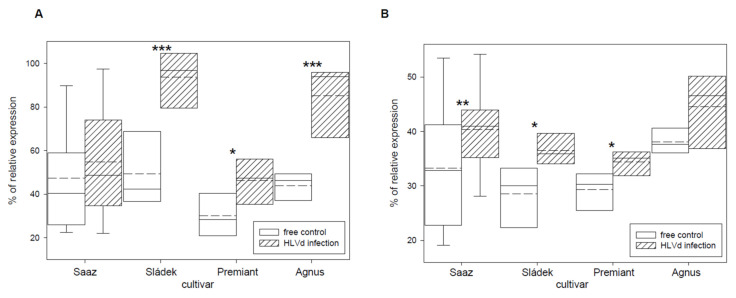
The percentage of the relative expression of alternatively spliced transcription factor IIIA (**A**) to the standard and alternative variant TFIIIA-7ZF, together (100%) detected by qRT-PCR, and (**B**) to the standard variant TFIIIA-7ZF, together (100%) detected by semiquantitative RT-PCR in young cones of HLVd-free and infected plants of hop cultivars. Probability level: *—*p* < 0.1, ******—*p* < 0.05, and ***—*p* < 0.01. Note: straight line—median; dashed line—average; and box—95% percentile ± standard deviation.

**Figure 8 plants-10-02297-f008:**
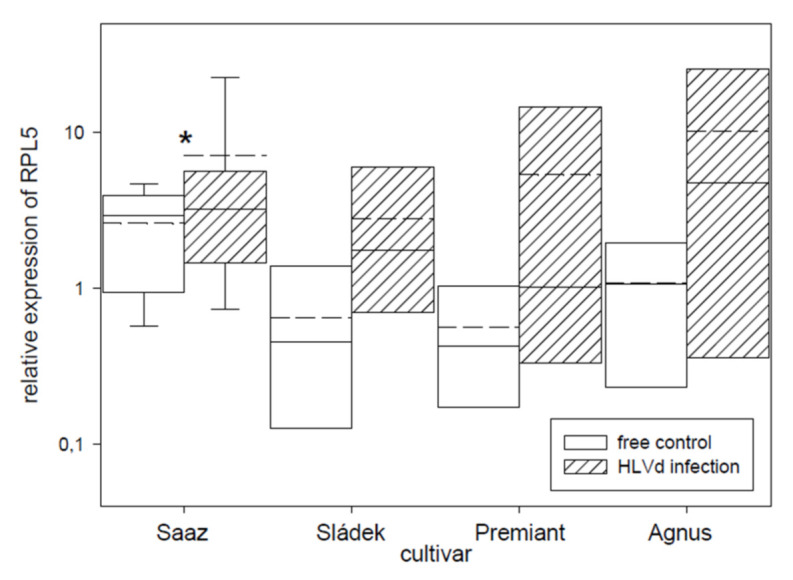
The relative expression to reference genes (RE = 1) for the RPL5 gene in young cones of HLVd-free and infected plants of hop cultivars. Probability level: *—*p* < 0.1, ******—*p* < 0.05, and ***—*p* < 0.01. Note: straight line—median; dashed line—average; and box—95% percentile ± standard deviation.

**Figure 9 plants-10-02297-f009:**
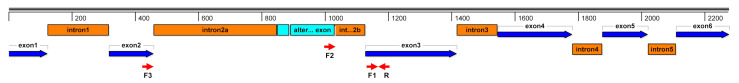
Genomic structure of transcription factor IIIA with PCR primers (red arrows) for alternative splicing (light-blue box) detection.

**Table 1 plants-10-02297-t001:** Hop bitter acids and xanthohumol contents (average ± SD) in dry cones of HLVd-free and infected plants of the Saaz cultivar.

HLVd Infection	Negative	Positive	Negative	Positive
**Harvest year**	**2019**	**2019**	**2020**	**2020**
**Number of samples**	**28**	**41**	**14**	**56**
Alpha acids (% of DW)	5.54 ± 0.63	3.84 ± 0.72 ***	5.46 ± 0.99	3.59 ± 0.60 ***
Beta acids (% of DW)	3.78 ± 0.32	3.96 ± 0.47 *	3.57 ± 0.34	3.63 ± 0.31
Cohumulone (% of AA)	21.34 ± 2.11	21.86 ± 2.50	20.07 ± 1.13	21.47 ± 2.13 **
Colupulone (% of BA)	43.07 ± 1.92	41.45 ± 3.07 **	38.79 ± 0.90	39.22 ± 2.28
Xanthohumol (% of DW)	0.309 ± 0.035	0.242 ± 0.040 ***	0.289 ± 0.032	0.221 ± 0.034 ***

Probability level: *—*p* < 0.1, ******—*p* < 0.05, and ***—*p* < 0.01. Abbreviations: DW—dry weight; AA—alpha acids; and BA—beta acids.

**Table 2 plants-10-02297-t002:** Hop bitter acids and xanthohumol contents (average ± S_D_) in dry cones of HLVd-free and infected plants of the Sládek, Premiant, and Agnus cultivars.

Cultivar	Sládek		Premiant		Agnus	
**HLVd infection**	**Negative**	**Positive**	**Negative**	**Positive**	**Negative**	**Positive**
**Number of samples**	**9**	**7**	**3**	**3**	**3**	**4**
Alpha acids (% of DW)	9.31 ± 0.98	6.68 ± 0.17 ***	8.86 ± 0.30	7.55 ± 0.34 ***	11.53 ± 0.45	10.52 ± 0.27 **
Beta acids (% of DW)	3.83 ± 0.46	4.44 ± 0.24 ***	3.32 ± 0.41	3.23 ± 0.33	3.90 ± 0.28	3.52 ± 0.20 *
Cohumulone (% of AA)	25.31 ± 0.55	23.63 ± 0.39 ***	17.13 ± 1.62	16.33 ± 0.51	29.97 ± 1.08	30.55 ± 0.61
Colupulone (% of BA)	49.21 ± 0.82	49.29 ± 0.47	37.30 ± 3.32	38.23 ± 2.12	52.80 ± 1.31	53.50 ± 0.57
Xanthohumol (% of DW)	0.606 ± 0.045	0.467 ± 0.027 ***	0.313 ± 0.021	0.247 ± 0.006 ***	0.747 ± 0.067	0.718 ± 0.058

Probability level: *—*p* < 0.1, ******—*p* < 0.05, and ***—*p* < 0.01. Abbreviations: DW—dry weight; AA—alpha acids; and BA—beta acids.

**Table 3 plants-10-02297-t003:** Contents of hop essential oils (average ± SD) in dry cones of HLVd-free and infected plants of the Saaz cultivar.

HLVd Infection	Negative	Positive	Negative	Positive
**Harvest year**	**2019**	**2019**	**2020**	**2020**
**Number of samples**	**21**	**6**	**14**	**56**
**Total oils (% of DW)**	0.527 ± 0.146	0.438 ± 0.108	0.443 ± 0.094	0.481 ± 0.112
**Monoterpenes (% of EO)**				
Myrcene	19.75 ± 3.12	24.60 ± 5.82 **	22.79 ± 3.71	25.26 ± 3.81 **
α-pinene	0.053 ± 0.013	0.083 ± 0.027 ***	0.082 ± 0.025	0.116 ± 0.026 ***
β-pinene	0.414 ± 0.070	0.603 ± 0.157 ***	0.615 ± 0.161	0.871 ± 0.186 ***
**Sesquiterpenes (% of EO)**				
α-humulene	20.72 ± 2.52	19.08 ± 2.27	20.38 ± 2.59	20.17 ± 2.67
β-caryophyllene	8.74 ± 1.00	7.02 ± 1.58 ***	7.09 ± 1.11	6.68 ± 1.03
β-farnesene	23.50 ± 3.00	20.79 ± 3.38 *	22.13 ± 3.00	17.96 ± 2.82 ***
γ-muurolene	1.098 ± 0.165	0.892 ± 0.263 **	0.796 ± 0.106	0.765 ± 0.120
β-bisabolene	0.409 ± 0.097	0.420 ± 0.040	0.474 ± 0.061	0.405 ± 0.093 ***
γ-cadinene	1.136 ± 0.172	0.912 ± 0.267 **	0.806 ± 0.107	0.779 ± 0.130
δ-cadinene	1.892 ± 0.257	1.512 ± 0.404 ***	1.295 ± 0.204	1.214 ± 0.201
Selinenes	1.626 ± 0.462	1.588 ± 1.086	1.108 ± 0.168	1.064 ± 0.244
**Alcohols (% of EO)**				
Linalool	0.201 ± 0.064	0.303 ± 0.084 ***	0.337 ± 0.124	0.554 ± 0.175 ***
Geraniol	0.035 ± 0.014	0.218 ± 0.127 ***	0.204 ± 0.278	0.514 ± 0.272 ***
**Epoxides ^A^ (% of EO)**	0.859 ± 0.483	0.995 ± 0.397	1.558 ± 0.522	3.070 ± 1.173 ***
**Ketones ^B^ (% of EO)**	2.637 ± 0.461	2.302 ± 0.289	2.838 ± 0.495	1.970 ± 0.483 ***
**Esters (% of EO)**				
Methylgeranate	0.082 ± 0.035	0.348 ± 0.150 ***	0.198 ± 0.139	0.672 ± 0.381 ***
Methylheptanoate	0.261 ± 0.087	0.259 ± 0.078	0.329 ± 0.090	0.434 ± 0.109 ***
Methyloctanoate	0.547 ± 0.204	0.432 ± 0.063	0.420 ± 0.163	0.314 ± 0.132 **
Methylnon-6-enoate	0.069 ± 0.030	0.093 ± 0.056	0.084 ± 0.035	0.157 ± 0.029 ***
Methyl-8-methylnonanoate	0.158 ± 0.053	0.121 ± 0.043	0.163 ± 0.046	0.114 ± 0.043 ***
Methyldeca-4,8-dienoate	0.433 ± 0.118	0.485 ± 0.113	0.409 ± 0.121	0.702 ± 0.126 ***
Methyldecanoate	0.383 ± 0.137	0.288 ± 0.045	0.259 ± 0.096	0.140 ± 0.081 ***
Methyldodeca-3,6-dienoate	0.924 ± 0.180	0.827 ± 0.070	0.813 ± 0.146	0.550 ± 0.173 ***

Probability level: *—*p* < 0.1, ******—*p* < 0.05, and ***—*p* < 0.01. Abbreviations: DW—dry weight; EO—essential oils; ^A^ Epoxides—sum of caryophyllene epoxide, humulene epoxide I a II; and ^B^ Ketones—sum of 2-nonanone, 2-decanone, 7-Me-2-decanone, 2-undecanone, and 2-tridecanone.

**Table 4 plants-10-02297-t004:** Contents of hop essential oils (average ± SD) in dry cones of HLVd-free and infected plants of the Sládek, Premiant, and Agnus cultivars.

Cultivar	Sládek		Premiant		Agnus	
**HLVd infection**	**Negative**	**Positive**	**Negative**	**Positive**	**Negative**	**Positive**
**Number of samples**	**7**	**4**	**2**	**2**	**2**	**2**
**Total oils (% of DW)**	1.836 ± 0.403	1.385 ± 0.130 *	1.270 ± 0481	1.085 ± 0.417	2.005 ± 0.813	1.775 ± 0.488
**Monoterpenes (% of EO)**						
Myrcene	34.06 ± 1.68	30.05 ± 5.60	29.65 ± 9.69	28.95 ± 4.03	32.35 ± 5.59	32.20 ± 5.37
α-pinene	0.087 ± 0.013	0.098 ± 0.025	0.070 ± 0.014	0.090 ± 0	0.165 ± 0.050	0.195 ± 0.007
β-pinene	0.677 ± 0.024	0.798 ± 0.139 **	0.595 ± 0.233	0.720 ± 0.113	1.055 ± 0.007	1.275 ± 0.304
**Sesquiterpenes (% of EO)**						
α-humulene	29.73 ± 1.27	31.50 ± 4.37	33.10 ± 7.21	33.65 ± 7.57	22.50 ± 0.57	20.55 ± 2.19
β-caryophyllene	12.20 ± 0.64	13.18 ± 2.15	10.90 ± 1.70	10.76 ± 1.48	13.80 ± 0.42	13.10 ± 1.41
β-farnesene	0.103 ± 0.119	0.093 ± 0.085	2.540 ± 0.255	1.885 ± 0.714	0.280 ± 0.071	0.185 ± 0.064
γ-muurolene	1.120 ± 0.180	1.113 ± 0.147	0.925 ± 0.085	0.800 ± 0	1.180 ± 0.170	1.220 ± 0.130
β-bisabolene	0.201 ± 0.172	0.048 ± 0.082	0.060 ± 0.060	0.165 ± 0.165	0	0
γ-cadinene	1.077 ± 0.039	1.153 ± 0.172	0.975 ± 0.105	1.153 ± 0.172	1.235 ± 0.145	1.250 ± 0.130
δ-cadinene	1.864 ± 0.109	2.018 ± 0.311	1.705 ± 0.205	1.570 ± 0.080	1.910 ± 0.170	1.830 ± 0.250
Selinenes	1.240 ± 0.247	1.158 ± 0.400	1.440 ± 0.240	1.100 ± 0.198	2.910 ± 0.655	2.940 ± 0.622
**Alcohols (% of EO)**						
Linalool	0.240 ± 0.028	0.310 ± 0.075 **	0.605 ± 0.262	0.600 ± 0.035	0.570 ± 0.113	0.625 ± 0.035
Geraniol	0.200 ± 0.064	0.343 ± 0.056 ***	0.050 ± 0.028	0.175 ± 0.021 **	0.740 ± 0.311	0.730 ± 0.325
**Epoxides^A^ (% of EO)**	0.547 ± 0.396	1.178 ± 0.553 *	0.280 ± 0.028	0.235 ± 0.035	2.315 ± 0.813	3.140 ± 0.354
**Ketones^B^ (% of EO)**	2.934 ± 0.632	2.330 ± 0.756	3.325 ± 0.813	1.990 ± 0.057	1.120 ± 0.057	1.180 ± 0.141
**Esters (% of EO)**						
Methylgeranate	0.330 ± 0.147	1.030 ± 0.301 ***	0.198 ± 0.139	0.405 ± 0.163	2.480 ± 0.368	2.950 ± 0.297
Methylheptanoate	0.407 ± 0.067	0.330 ± 0.149	0.395 ± 0.064	0.355 ± 0.064	0.190 ± 0.028	0.225 ± 0.212
Methyloctanoate	0.837 ± 0.238	0.618 ± 0.303	0.315 ± 0.021	0.295 ± 0.177	0.225 ± 0.021	0.260 ± 0.071
Methylnon-6-enoate	0.044 ± 0.013	0.058 ± 0.015	0.060 ± 0.028	0.080 ± 0.028	0.040 ± 0.014	0.040 ± 0.014
Methyl-8-methylnonanoate	0.084 ± 0.026	0.065 ± 0.033	0.110 ± 0.014	0.060 ± 0 **	0.195 ± 0.007	0.200 ± 0.014
Methyldeca-4,8-dienoate	0.513 ± 0.089	0.628 ± 0.213	0.570 ± 0.255	0.865 ± 0.431	0.340 ± 0.057	0.390 ± 0.127
Methyldecanoate	0.410 ± 0.127	0.270 ± 0.123	0.245 ± 0.021	0.160 ± 0.071	0.125 ± 0.035	0.150 ± 0.071
Methyldodeca-3,6-dienoate	0.390 ± 0.288	0.145 ± 0.290	0.285 ± 0.403	0.265 ± 0.375	0	0

Probability level: *—*p* < 0.1, ******—*p* < 0.05, and ***—*p* < 0.01. Abbreviations: DW—dry weight; EO—essential oils; ^A^ Epoxides—sum of caryophyllene epoxide, humulene epoxide I a II; and ^B^ Ketones—sum of 2-nonanone, 2-decanone, 7-Me-2-decanone, 2-undecanone, and 2-tridecanone.

**Table 5 plants-10-02297-t005:** List of analyzed sequences of secondary metabolites biosynthesis genes, transcription factors, and reference genes.

Abbrev.	Gene	Number *
	Bitter acids biosynthesis	
BCAT1	Branched-chain amino acid aminotransferase 1	002627F.g2, JQ063073
VPS	Phloroisovalerophenone synthase	001329F.g74, FJ554588
PT1L	2-acylphloroglucinol 4-prenyltransferase	KM222441
PT2	2-acyl-4-prenylphloroglucinol 6-prenyltransferase	KM222442
HS1	Monooxygenase 2 (Humulone synthase 1)	010625F.g1, 008956F.g7, KJ398144
HS2	Monooxygenase 2 (Humulone synthase 2)	008118F.g14, KJ398145
	Polyphenols and flavonoids biosynthesis	
	Flavonoids biosynthesis	
PAL	Phenylalanine ammonia-lyase	000198F.g108, g113, g114, g115, g116, g117
4CL2	4-coumarate-CoA ligase 2	001395F.g25
CHSH1	Naringenin-chalcone synthase (CHS_H1)	000203F.g58, AJ304877
OMT1	O-methyltransferase 1 (Desmethylxanthohumol 6’-O-methyltransferase)	000009F.g116, EU309725
F3H	Flavanone 3-hydroxylase	001909F.g33
LAR	Leucoanthocyanidin reductase	000109F.g58, HQ734722
	Terpenes biosynthesis	
GPPS-SSU	Geranyl diphosphate synthase small subunit	001483F.g7, FJ455406
FPPS	Farnesyl pyrophosphate synthase	000817F.g23, AB053487, AF268889
MTS1	Monoterpene synthase 1	000149F.g29, EU760348
MTS2	Monoterpene synthase 2 (myrcene synthase)	003722F.g32, EU760349
STS1	Sesquiterpene synthase 1 (α-humulene synthase)	001011F.g31, EU760350
STS2	Sesquiterpene synthase 2 (germacrene-A synthase)	001011F.g31, EU760351
TPS9	Terpene synthase 9	001370F.g12
NES	(E)-nerolidol/linalool synthase	004063F.g13
	Transcription regulation of biosynthesis genes	
MYB3	Transcription factor HlMYB3	AM501509
MYB8	Transcription factor HlMYB8	002031F.g25, HG983335
MYB78	Transcription factor MYB78	000063F.g63
bHLH2	Transcription factor HlbHLH2 (TT8)	000662F.g4, FR751553
bHLH4	Transcription factor GLABRA 3 (HlbHLH4)	001145F.g21, HG983336
WRKY1	WRKY transcription factor 1	000029F.g2, CBY88801
TFIIIA	Transcription factor IIIA	002165F.g6
RPL5	Ribosomal protein L5	GAAW01025872
	Reference genes	
TTG1	Protein TRANSPARENT TESTA GLABRA 1 (HlWD40)	002162F.g15, FN689721
MYC2	Transcription factor MYC2	001862F.g5
PIF4	Transcription factor PIF4	000802F.g1
GAPDH	Glyceraldehyde-3-phosphate dehydrogenase	004935F.g2, 004935F.g5
RH46	DEAD-box ATP-dependent RNA helicase 46	000004F.g75

* HopBase gene number or NCBI GenBank accession number.

**Table 6 plants-10-02297-t006:** Positions of plant cis-acting regulatory DNA elements on promoter sequences before ATG-start of secondary metabolites biosynthesis genes.

cis elements	MYB1AT	MYB2CONSENSUSAT	MYBCORE	MYB1LEPR	MYBCOREATCYCB1	MYBPZM	BOXLCOREDCPAL	MYBPLANT	MYBST1	MYBGAHV	MYCCONSENSUSAT	WRKY71OS
**Sequence/gene**	WAACCA	YAACKG	CNGTTR	GTTAGTT	AACGG	CCWACC	ACCWWCC	MACCWAMC	GGATA	TAACAAA	CANNTG	TGAC
BCAT1	−439					−1688					−1745, −1089, −925, −792, −695	−1643, −1093, −890, −584
VPS	−1559	−1924, −1914, −1826, −1733, −1715, −1702, −474	−1873, −1236, −874	−1865	−1845, −1074	−1554	−1555	−1556, −1473, −1431	−1020	−996	−1924, −1826, −1770, −1733, −1702, −1488, −1448, −1316, −811, −474, −77	−1920, −1886, −1852, −1815, −1689, −1679, −1676, −961, −470
PT1L	−512		−1536, −1313		−1334, −1153	−1801, −401, −397, −205	−398	−399	−1636, −1488, −769	−1106	−1611, −1546, −1408, −1325, −333	−1987, −1698, −1507, −1420, −1057, −980, −928, −638, −597, −558
PT2	−466	−1810, −1331, −271	−27								−902, −350, −271, −8	−1891, −1634, −1001, −777, −510, −345, −340
HS1	−126		−1907	−1905	−953	−431	−432	−433			−1609, −1526, −1509, −1433, 537, −490, −425, −224, −187	−1959, −1917, −1860, −647, −594, −495, −280
HS2	−120		−349			−1612				−137	−1809, −1730, −1714, 1624, −1423, −600, −538	−921, −814, −734, −605, −427, −398, −345, −243
PAL	−830, −539, −391, −321, −143, −116		−1242, −665, −308			−919, −492	−493	−494	−1450	−1998, −789	−1979, −1727, −1259, −1211, −894	−1648, −1265, −1024, −835, −577, −414, −409, −333, −237, −233, −176
4CL2	−937, −726, −361	−1896, −1790	−1877, −1694, −183			−169, −118	−170, −119		−1238, −1112	−1146	−694, −547, −497, −300, −144	−1438, −1011, −800
CHSH1	−1800	−837, −476	−1526, −385	−622	−836, −475		−1427, −517		−1861	−245	−1303, −1042, −224	−1665, −1640, −1283, 1179, 1148, −725, −284, −220
OMT1	−1769, −556										−1496, −1306	−1034, −393
F3H	−1499, −141, −134, −110					−1441				−1544, −210	−1934, −1538, −1468, −881, −734, −632, −397	−1937, −1634, −1534, −1277, −330, −256
LAR	−978, −390		−1948, −1711, −1580	−1578					−1723, −297		−1967, −1877, −1267, −974, −830, −82	−650, −609, −328
GPPS-SSU	−1905, −1870, −1788, −1515, −1222, −270, −8		−1984, −30	−1750	−635	−1736, −1296	−1297	−1787, −1298	−748	−625	−1987, −1766, −1495, −1344, −727, −571, −366, −305, −189, −49	−1947, −1693, −1491, −805, −723
FPPS			−26								−1842, −1092, −772, −404, −174	−1779, −1698, −1661, −1337, −595, −119
MTS1	−1305		−193, −173						−1305, −1098		−1253, −845, −386, −117	−1930
MTS2	−905, −566					−1796		−1193	−1993	−1194	−1933, −399	−1407, −1345, −978, −836, −708, −657, −536, −312
STS1	−1196, −455					−1875			−983		−810, −733, −531, −471, −65	−1407, −1090, −640, −527, −498, −297
STS2	−1212, −452					−1892			−995		−807, −730, −528, −468, −65	−1423, −1105, −637, −524, −495, −294
TPS9	−1909, −283		−1160, −12		−18						−1273, −585, −12	−1814, −1555, −1305, −1257
NES	−618	−929, −692	−545, −500, −327		−928		−1411		−834	−1634	−1865, −1796, −1539, −1476, −1101, −692, −500, −462, −310	−1286, −1210, −788, −688, −313, −100

## Data Availability

Not applicable.

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
