# Peer review of "The Influence of Hop Latent Viroid (HLVd) Infection on Gene Expression and Secondary Metabolite Contents in Hop (Humulus lupulus L.) Glandular Trichomes"

_plants, 2021, doi:10.3390/plants10112297_

Round 1

Reviewer 1 Report

In their manuscript (plants-1400834) titled "The influence of hop latent viroid (HLVd) infection on genes expression and secondary metabolites contents in hop (Humulus lupulus L.) glandular trichomes", Josef Patzak, Alena Henychová, Karel Krofta, Petr Svoboda and Ivana Malířová evaluate changes to candidate metabolite and gene accumulation levels upon HLVd infection, in mature hop cones. They analyzed several cultivars for alpha and beta acids, terpenes and essential oils. Then they looked for differential transcript accumulation of genes involved in the different biosynthetic pathways; either enzyme coding genes or genes coding for upstream regulators. Next, they investigated the accumulation of differently spliced mRNAs of the transcription factor TFIIIA, whose splicing variant TFIII-7ZF was recently identified as essential for Pol II to replicate PSTVd, the type member of the family Pospiviroidae. Finally, they measured the transcript accumulation of RPL5, which was shown to be involved in the splicing regulation of TFIIIA mRNA.

I find this work interesting and adequate for publication in Plants (special issue Plants Viroid/Viruses: Insight into Genome and Epidemiology) because it advances our knowledge on the impacts of so-called "symptomless" viroids on crop plants, however it needs minor modifications to meet my concerns.

Major concerns:

  • The introduction describes the known changes in hop secondary metabolites upon viroid infection in general. However, it should better explain why mono- and sesquiterpenes, epoxides, ketones, essential oils, alpha & beta acids and so on, are important to monitor in hops, before describing the effect of HLVd infection on their accumulation levels.
  • Also, the last part of the introduction should stress the novelty of this work.
  • The results section should start with a sentence explaining that the infectious status of each sample (either HLVd-free or HLVd-infected) was checked.
  • The different findings (both in the result and discussion sections) are presented as a catalog; one would expect a synthesis of the results, what is common to all cultivar? what varies? and more importantly how pertinent these differences/similarities are in regard to known effects of infection on cone quantities and qualities.
  • The authors present the ratio of differently spliced RNAs of TFIIIA but it is not clear from the results whether the overall accumulation level is changed or not. This could be added either in the figures with other TF genes (Fig 7) or with the splicing variants (Fig 8). A short explanation of what is known about this TF and RPL5 in the replication of PSTVd is needed. Is RPL5 mRNA increased upon PSTVd infection? The authors conclude that the results are similar for HLVd infection, although the increase of RPL5 mRNA accumulation they observe is not statistically significant in most cultivars. This should at least be discussed.
  • The conclusion about the involvement of TFIIIA and RPL5 in HLVd should be changed. This work does not formally demonstrate that the transcription factor (or its regulatory cascade) is involved (line 29), and even less, necessary for HLVd replication (lines 26). Only an increase (not statistically significant for RPL5) of their transcripts is shown. Please rephrase to avoid over conclusion.

Additional comments:

  • Introduction: a better description of the viroids found in hop is needed, this should include the genus they belong to. Please specify that citrus bark cracking viroid (CBCVd) and hop latent viroid (HLVd) both belong to the genus Cocadviroid whereas hop stunt virus is a member of the genus Hostuviroid and apple fruit crinkle viroid (AFCVd) is related to the genus Apscarviroid.
  • Line 12: replace "from Pospiviroidae family" by "of the family Pospiviroidae".
  • Line 13 (and throughout the manuscript): virus & viroid names should not have a capital letter nor be italicized, except when referring to taxonomy (see ICTV recommendations) => replace "Hop latent viroid" with "Hop latent viroid" at the beginning of a sentence or "hop latent viroid" within sentences.
  • Lines 14-15: "As producers of virus and viroid free hop mericlones, we evaluated the influence of HLVd infection on the content…". Please clarify the rationale, or remove the first part of the sentence. Indeed, to produce virus- and viroid-free mericlones, it is sufficient to check for the absence of virus and viroid.
  • Line 38: change "Pospiviroidae are replicated" to "Viroids of the family Pospiviroidae" as viruses and viroids but not a taxon can be replicated. Alternatively, you could possibly write "pospiviroids are replicated"
  • Line 40: change families to family.
  • Line 44: please specify what is an alpha-acid (hydroxyl function in position alpha of the carbonyl group).
  • Line 49: make clear that the bitter acids are or derive from alpha acids.
  • Line 53: change increment to increase. This also applies line 237.
  • Line 60: remove an
  • Line 65 and throughout the manuscript: replace "reduced from4% to 29%" with "reduced by 4.4% to 29%"
  • Line 70: it is not clear what "was vice versa lower" means.
  • Line 77: should it not rather be pathways (plural)?
  • Lines 82-85: it is not quite clear what differentiates vsRNA and viroid-induced RNAi as pathogenesis mechanisms. Please specify, as small RNAs are not limited to TGS and it might not be obvious that RNAi is synonymous to PTGS. Also, direct interaction between the viroid and plants proteins could be exposed in another sentence.
  • Line 86 It is not clear what vd-sRNA stand for as the abbreviation given line 83 for viroid small RNAs is vs-RNAs. Also, vs-sRNA should not appear twice in this sentence.
  • Line 89: remove "an".
  • Line 101: the abbreviation of hop stunt viroid is already defined and should be used here alone. The same applies for the 2 other viroids cited lines 102 & 103.
  • Table 3: AA and BA do not appear in the table thus remove from the legend. Indicate what statistical tests has been used to calculate P. Control should be preferred to negative. Control samples represent the reference; thus, one would expect that the significant differences are shown on the positive (HLVd-infected) samples. This applies to other figures and tables too.
  • Line 123: is Figure 2 meant?
  • Line 142 & table 5: is α- and β- pinene contents meant?
  • Line 214: If WRKY is indeed significantly downregulated in cones of cultivar Premiant it should appear with star(s) in panel 7F. Please check.
  • Lines 225-226: " Ribosomal protein L5 (RPL5) is also involved in TFIIIA transcription regulation". Please be more precise, and cite reference.
  • Table 2: as the sequences described in this table are not further investigated/used in the study, the table could be either moved to supplementary material or even removed.
  • Line 220: It is important to cite a reference for TFIIIA, here.
  • Line 221: Is it really the electrophoresis that is semi-quantitive?
  • Fig 8: What is the rationale for a sqRT-PCR analysis in addition to the RT-qPCR?
  • Fig 8 caption and line 396: please clarify whether the results present the expression of alternatively spliced mRNAs compared to standard mRNAs as mentioned in caption of fig 8 or relatively to all variants as mentioned line 396. Also clarify whether it is incorrect or alternative splicing.
  • Line 269: correct prenyl
  • Line 298: change "gene's expression changes" to "gene expression changes".
  • Line 314: change presented to present.
  • Line 404: replace "measured by pGEM DNA…" with "estimated by comparison with pGEM DNA…"
  • Line 423: formally, the TFIIIA-7ZF regulatory factor is not spliced, unlike its mRNA. Please rephrase.

And throughout the manuscript:

  • Check for writing of viroid names: remove capitals and italics.
  • Check for "reduced by x% to y %.
  • Renumber tables and figures in the order they appear in the text.
  • Tables should have a title appearing above the table, all other indications should appear below the table.
  • All figures: Title and legend should appear below the figure, not above.
  • Fig 2 to Fig 9: replace letters with cultivar names and give a title = gene name in the box plots. This will help grasp graphical representation of the results and will also remove ambiguity between panel naming and data in x-axis.
  • Make the difference between the gene and the protein/enzyme it codes for. As an example: "gene GPPS-SSU, a part of the heterodimeric enzyme complex…" or both genes MTS1 and MTS2, enzymes for monoterpene biosynthesis…"
  • Fig 4 to Fig 9: results are relative expression, please specify to what they are relative. What expression is set to 1 (Fig 4-8) or to 100 % (Fig 9).
  • Consider "to depend on" instead of "be depend on" (e.g. lines 148, 149, 158, 273).
  • It is not clear what the authors mean by "join with" (e.g. lines 219, 279).
  • Change "is genetically dependent" to "is genotype dependent" (e.g. lines 55, 68, 299).
  • Once an abbreviation is defined, use it. The viroid name should not appear in full throughout the entire text (e.g. lines 231, 336, 411, 426).

References:

  • Remove extra capital letters in reference titles (e.g. ref 5, 7, 17…)

Author Response

We accepted all of recommendations and made proposed corrections.

Reviewer 2 Report

The manuscript “The influence of Hop latent viroid (HLVd) infection on genes expression and secondary metabolites contents in hop (Humulus lupulus L.) glandular trichomes” concerns changes in the activity of genes related to the synthesis of secondary metabolites under biotic stress in hop. The gene activity has been set together with the compound content.

The presented results are interesting and the manuscript is properly written. However, I have some little remarks.

Lines 18-20. There is  “We confirmed that viroid infection significantly reduced the contents of alpha bitter acids from 8.8% to 34%. New, we found that viroid infection significantly reduced the contents of xanthohumol from 3.9% to 23.5%.”

I have some doubts about the phrase “… reduced … from 8.8% to 34%” and “… reduced from 3.9% to 23.5%.” I guess that the authors mean range of reduction (the least reduction – by 8.8% and the largest - by 34%). But the sentence construction suggests that there was 8.8% and it reduced to 34% which sounds illogical. Therefore, the authors should rewrite this sentence to make it clear.

Line 65. As above.

Line 69. “-pinene”?

Line 123 and 131. The content of alpha bitter acids is presented on figure 2.

Line 142. As in line 69. There is lack of something. – and –pinene?

Line 142. As in lines 18-20. …varied from 32.8% to 56.6%? or by 32.8% to by 56.6%?

Line 180-182. The sentence “From these genes, PAL (Figure 4A) and CHSH1 180 (Figure 4C) genes were only up-regulated by infection in cones of Premiant (5.7 times) and 181 Saaz (2.6 times) cultivars, respectively.” is difficult to understand. The authors try to include too much information in one sentence.

Line 207. 16 times? Or 1.6 times?

Line 214-215. “Because, there are unknown...” this phrase is unclear.

Line 344. The HLVd infected plants were obtained from infected field plants or were infected after in vitro regeneration from healthy plants? Please clear it.

Line 387. What was the concentration of cDNA?

Figures are well explained but they would be more friendly for a reader. I suggest to write names of cultivars on X-axis and the name of a tested gene in the title of Y-axis i.e. “relative expression of VPS gene”.

Author Response

(The authors gave the same response as above.)
